# Digital spatial profiling of segmental outflow regions in trabecular meshwork reveals a role for ADAM15

**Jennifer A. Faralli**[1], **Mark S. Filla**[1], **Yong-Feng Yang**[2], **Ying Ying Sun**[2], **Kassidy Johns**[1], **Kate E. Keller**[2], **Donna M. Peters**[1,3]*

1 Departments of Pathology & Laboratory Medicine, University of Wisconsin, Madison, Wisconsin, United States of America, 2 Casey Eye Institute, Oregon Health & Science University, Portland, Oregon, United States of America, 3 Ophthalmology & Visual Sciences, University of Wisconsin, Madison, Wisconsin, United States of America

* dmpeter2@wisc.edu

**Data Availability Statement:** All relevant data are within the manuscript and its Supporting information files.

## Abstract

In this study we used a spatial transcriptomics approach to identify genes specifically associated with either high or low outflow regions in the trabecular meshwork (TM) that could potentially affect aqueous humor outflow in vivo. High and low outflow regions were identified and isolated from organ cultured human anterior segments perfused with fluorescently-labeled 200 nm FluoSpheres. The NanoString GeoMx Digital Spatial Profiler (DSP) platform was then used to identified genes in the paraffin embedded tissue sections from within those regions. These transcriptome analyses revealed that 16 genes were statistically upregulated in high outflow regions and 57 genes were statistically downregulated in high outflow regions when compared to low outflow regions. Gene ontology enrichment analysis indicated that the top three biological categories of these differentially expressed genes were ECM/cell adhesion, signal transduction, and transcription. The ECM/cell adhesion genes that showed the largest differential expression (Log2FC ±1.5) were *ADAM15*, *BGN*, *LDB3*, and *CRKL*. *ADAM15*, which is a metalloproteinase that can bind integrins, was upregulated in high outflow regions, while the proteoglycan *BGN* and two genes associated with integrin signaling (*LDB3*, and *CRKL)* were downregulated. Immunolabeling studies supported the differential expression of ADAM15 and showed that it was specifically upregulated in high outflow regions along the inner wall of Schlemm's canal and in the juxtacanalicular (JCT) region of the TM. In addition to these genes, the studies showed that genes for decorin, a small leucine-rich proteoglycan, and the α8 integrin subunit were enriched in high outflow regions. These studies identify several novel genes that could be involved in segmental outflow, thus demonstrating that digital spatial profiling could be a useful approach for understanding segmental flow through the TM. Furthermore, this study suggests that changes in the expression of genes involved in regulating the activity and/or organization of the ECM and integrins in the TM are likely to be key players in segmental outflow.

**Funding:** This study was supported by grants EY017006, EY032905 (DMP) and EY032590, EY019643 (KK) and Core grants P30 EY01665 (UW-Dept. of Ophthalmology) and P30 EY010572 (Dept. of Ophthalmology and Casey Eye Institute).

**Competing interests:** The authors have declared that no competing interests exist.

## Introduction

Intraocular pressure (IOP) is maintained by the balance between the level of aqueous humor produced by the ciliary body and the rate by which it exits the eye through the trabecular meshwork (TM) in the anterior portion of the eye [1–4]. The movement of aqueous humor outflow is considered to be segmental, or non-uniform, around the circumference of the eye so that regions of high flow are intermingled with regions of low and medium flow [5–8].

What controls segmental outflow, however, remains to be determined. Regions of low and high outflow in the TM appear to be morphologically similar, although some differences have been noted that could explain segmental outflow. Regions of high outflow have been reported to be associated with collector channel ostia and distal vasculature [9, 10]. These high outflow regions often have a thicker, expanded TM [11] and contain fewer connections between the cells lining the inner wall of Schlemm's canal (SC) and the underlying JCT. More basal openings below giant vacuoles [12] have also been observed in high outflow areas. Interestingly, the distribution of these regions appears to be dynamic and age-dependent with older eyes exhibiting a slower redistribution then younger eyes [13, 14].

At the cellular level, both the composition of the extracellular matrix (ECM) and the cells residing in the JCT and underneath the adjoining inner wall endothelium of SC are thought to contribute to the movement of aqueous humor outflow across the TM [15–19]. Studies using transgenic mouse models and cultured anterior segments have indicated that changes in the expression of a variety of ECM proteins, including fibronectin [20, 21], collagens [22], SPARC [23, 24], thrombospondins [25], decorin [26], and myocilin [27, 28], affect outflow. ECM PCR arrays and proteomic studies further supported this idea and suggested that expression of some ECM genes and proteins, such as versican, VCAM, and TIMP1, appear to be enriched in certain regions of the TM relative to outflow [29–32].

Studies using cultured models of anterior segments have also shown that the biomechanical properties of the low and high outflow regions differ [30] and that low outflow regions appear to be regions where the tissue is stiffer. Subsequent proteomic studies on these tissues have found that significant differences in ECM proteins existed when the anterior segments were perfused at a higher pressure [30]. Among the proteins identified were several proteoglycans (keratocan, decorin, and lumican) that play a role in collagen fibril formation and the ECM proteins, dermatopontin and thrombospondin-4. Transforming growth factor β1 (TGFβ1) and MAM Domain Containing 2 (MAMDC2) were also identified in this study. Subsequent studies using TM cells isolated from these high and low outflow regions have also identified differences in the ECM proteins expressed by these cells but only after a prolonged time in culture [33]. Studies have also shown that variations in TM cellularity in high and low outflow regions do exist, but these differences did not appear to be correlated with segmental outflow [34]. Although the molecular differences identified in these segmental areas are likely to be correlated with differential outflow facility across different areas of the TM circumference [35], the molecular mechanism(s) that control normal aqueous humor outflow are still poorly understood.

In this study, we used digital spatial profiling of the transcriptomes of high and low outflow regions of human anterior segments to determine if differential expression of ECM or cell adhesion genes can account for segmental outflow. This study shows that relatively few structural ECM and integrin genes, with the exception of *DCN* and the *ITGA8* integrin subunit, showed any differential expression between low and high outflow regions. Rather, the changes observed were mainly in genes such as *ADAM15*, *CRKL*, and *LDB3* that regulate cell adhesion, especially adhesion involving integrins and most of the changes were less than 3-fold. This suggests that large changes in gene expression may not be involved in segmental outflow and that

outflow could be regulated by small changes in gene expression that control the activity of the ECM and cell adhesion molecules.

## Materials and methods

### Donor eyes

Normal human cadaver eyes were obtained from 3 males (ages 65–70). One male was Caucasian. The race of the other two donors was unknown. All eyes were acquired from Lions-Vision Gift, Portland, OR, and none of the eyes had a history of ocular disease. All studies were approved by the Oregon Health and Science University Institutional Review Board and were done in accordance with the tenets of the Declaration of Helsinki for the use of human cadaver tissue.

### Anterior segment perfusion culture

Human cadaver eye globes (n = 3) were bisected and the lens, ciliary body, and iris were removed leaving the cornea, TM, and a small rim of sclera. Anterior segments from each donor eye then were placed in stationary organ culture as previously described [36]. Length of time from death to stationary culture was less than 48 hours and anterior segments were initially placed into serum-free stationary organ culture for 5–7 days to facilitate postmortem recovery. Anterior segments were then perfused with serum-free Dulbecco's Modified Eagle's Medium (a 1:1 mixture of high and low glucose DMEM) containing 1% Penicillin/Streptomycin/Fungizone until a stable baseline was achieved. Perfusion was done at a constant pressure (8.8 mmHg) with an average flow rate of approximately 1–7 μl/min, which is similar to normal physiological rate and pressures (minus episcleral venous pressure) in vivo [13, 30, 31, 33, 37].

Regions of segmental outflow were then determined by perfusing anterior segments for 1 hour with fluorescently-labeled amine-modified 200 nm FluoSpheres as described previously [31]. The distribution of fluorescently tagged FluoSpheres was imaged by fluorescence microscopy and wedges of anterior segments containing regions of high fluorescence intensity were dissected away from regions of medium and low fluorescence intensity (Fig 1A). Anterior segment tissue wedges containing the TM from high, medium and low outflow regions were then fixed with 4% paraformaldehyde in PBS for 30 min at room temperature. Tissues were then placed in 70% ethanol, shipped to University of Wisconsin (UW), and embedded into paraffin at the UW-Translational Research Initiatives in Pathology histopathology core facility.

### NanoString digital spatial profiling (DSP)

Wedges of anterior segments from low and high outflow regions were prepared for DSP analysis as they would be expected to show the biggest differences in gene expression. Briefly, paraformaldehyde-fixed paraffin sections (5 μm) from the designated high and low outflow regions from the three eyes were cut, placed on a single SuperFrost Plus slide, and deparaffinized by baking the sections for 10 min at 60°C followed by two 5 min xylene washes, a graded series of ethanol washes (100–50%), one 5 min nuclease-free water wash and two 5 min PBS washes. Antigen retrieval was performed by incubating the sections for 15 min at room temperature in 10 mM EDTA with 0.1 M Tris buffer (EDTA/Tris) at pH 9.0 (Invitrogen 00-4956-58) and then incubating the sections in 95°C EDTA/Tris buffer for 20 min. Sections were allowed to gradually cool to room temperature for 20 min and then washed with two 5 min washes of TBS-0.05% Tween-20, pH 7.4. Sections were then incubated for 15 min at 37°C with 1μg/ml proteinase K in PBS and then washed with PBS. This was followed by one 5 min wash with 10% neutral buffered formalin, two 5 min washes with NBF stop buffer and one wash with PBS.

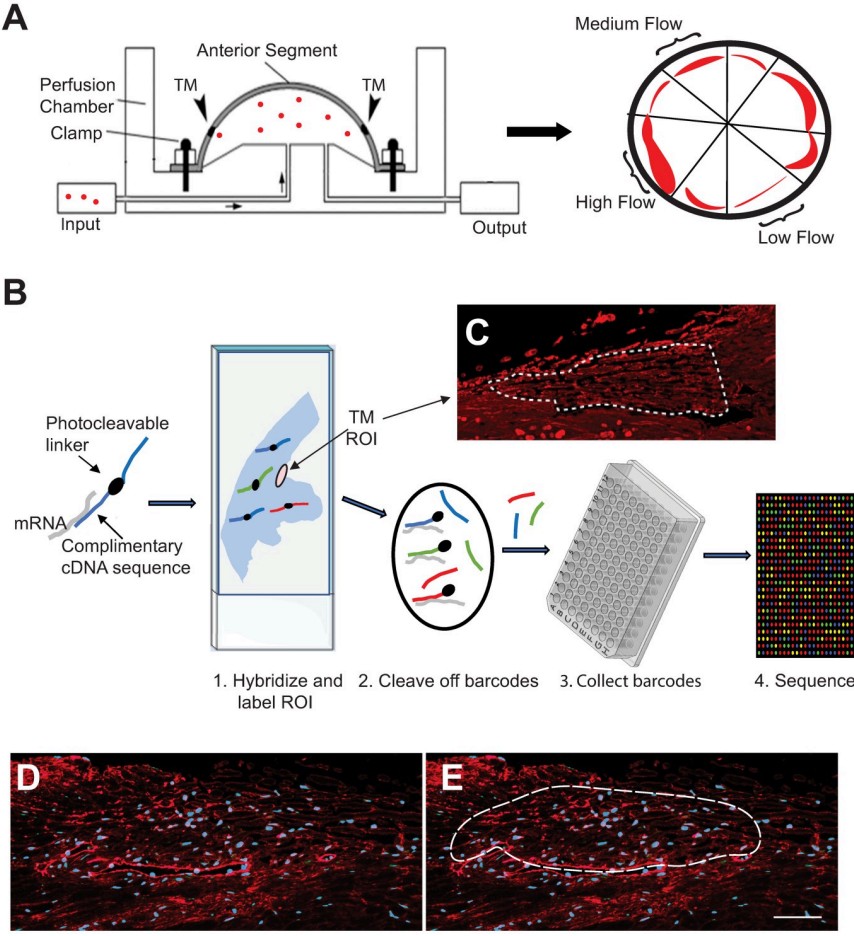

**Fig 1. Schematic of the workflow for the study.** (A) Schematic of organ culture system. Fluorescent beads (red dots) are perfused through a human anterior segment in organ culture. After 1 hour the anterior segment is divided into low, medium and high outflow regions based on the intensity of the fluorescent beads (red regions). (B) Images show steps in the DSP process. Tissue sections on a slide were labeled with DSP probes and ROIs were selected based on tissue morphology and fibronectin immunostaining. Barcodes in ROIs were cleaved using UV light and collected into a 96-well plate for sequencing. (C) Actual scan of fibronectin labeled tissue from a high outflow region with the ROI drawn. (D-E) Images of fibronectin (red) labeled tissue sections. Dashed line in panel E shows a typical size and location of the ROI used for the study. Nuclei (blue) were labeled with Hoechst 33342.

Sections were then hybridized overnight with the GeoMx Human Whole Transcriptome Atlas (WTA) oligonucleotide probe mix (NanoString Technologies, Inc) according to the manufacturer's instructions. Each RNA probe was coupled with a unique photo-cleavable oligonucleotide barcode (Fig 1B). The tissue was then double-labeled with an anti-fibronectin antibody conjugated to Alex 647 (1:200 dilution Abcam 198934) and Syto13 (1:10 dilution Thermo NanoString GeoMx Nuclear Stain Morphology Kit 121300303). Labeled slides were imaged on the GeoMx® digital spatial profiler and the TM was identified visually by its unique morphological structure and the fluorescent labeling pattern of fibronectin. For each TM identified, a single polygon-shaped region of interest (ROI) ranging in size from 14,000–40,000 $\mu m^2$ was drawn around the TM encompassing the inner wall (IW) of SC, the JCT and the trabecular beams (Fig 1C). Each ROI was then illuminated with a UV light so the oligonucleotides probe barcodes present within each TM were photocleaved and collected individually into a single well of a 96-well collection plate. Barcodes were sequenced at the UW

Biotechnology Center on an Illumina NovaSeq 6000 and FASTQ files were then processed by the NanoString pipeline. Statistical analysis of the data was performed using a linear mixed model and the Benjamini-Hochberg correction factor to control for false discovery rate. Digital counts were normalized with internal controls for system variation.

### Immunohistochemistry of high and low flow regions

Five-micrometer paraffin sections from high and low outflow wedges mounted onto glass slides were deparaffinized in xylenes followed by rehydration in a series of 100–50% ethanol solutions. Antigen retrieval was performed at 95˚C in 10 mM EDTA with 0.1 M Tris buffer at pH 9.0 for 20 min. The sections were allowed to cool to room temperature and were blocked for 1 hour with 1% BSA in PBS. Sections were then labeled with rabbit monoclonal anti-ADAM15 antibody (Abcam ab124698) diluted 1/800 in 1% BSA in PBS overnight at 4˚C. Sections were then washed with PBS and labeled with a secondary Alexa 546-conjugated goat anti-rabbit IgG (A11035; ThermoFisher Scientific) diluted 1/500 in the 1% BSA/PBS buffer. Nuclei were labeled with Hoechst 33342. Sections were washed with PBS and mounted with a glass coverslip using Shandon™ Immu-mount (ThermoFisher Scientific). Labeled cells were imaged using a Zeiss Imager.M2 fluorescence microscope together with Zen image acquisition software ver 3.079. Images of ADAM15 labeling were equally thresholded using Image J (National Institutes of Health) and the integrated density was calculated for the JCT and IW combined, the outer wall (OW) and beams of each image. High outflow (n = 11 wedges from 3 OD eyes) and low outflow (n = 9 wedges from 3 OD eyes) regions were averaged and statistical analysis was performed using ANOVA in conjunction with Tukey's HSD test.

## Results

To get a better idea of how outflow facility may be regulated at the gene level, we used the NanoString GeoMx platform (Fig 1B) to compare the transcriptome of high and low outflow regions. The TM from 10 regions of high outflow and 13 regions of low outflow were selected as described in Materials and Methods. Not surprisingly, the transcriptomes of the two regions were very similar and only 74 genes were found to be statistically different (p<0.05). The majority of these genes (57 genes) were downregulated in high outflow regions compared to the low outflow regions and 16 genes were upregulated in the high outflow regions compared to low outflow regions (Fig 2). Somewhat surprisingly, only 10 genes showed a log2FC of ±1.5 or greater suggesting that large differences in gene expression were not associated with segmental outflow in the TM.

To identify biological processes that may be affected by the genes that were statistically different, gene ontology analysis was performed. Fig 3 shows that the biological classification involving the greatest number of genes which were differentially expressed between high and low outflow regions was associated with the ECM and cell adhesion. As shown in Table 1, most of the genes appeared to regulate cell adhesion or ECM organization and, with the exception of *ADAM15*, were all downregulated in high outflow regions. The greatest differences appeared to involve the genes *CRKL* and *LDB3* which encode for proteins found in focal adhesions that would affect integrin signaling. Both *LDB3* and *CRKL* showed a log2FC = -1.6 (or a 0.3-fold decrease) in expression. The next most downregulated gene was *BGN* (log2FC = -1.4), which showed a 0.4-fold decrease in expression. *ADAM15* (log2FC = 0.7), the only gene upregulated in the high outflow region, showed a 1.6-fold increase in expression.

Fig 4 shows the boxplots for the distribution of the *ADAM15*, *LDB3*, *CRKL* and *BGN* in high and low outflow regions of the TM. The majority of the values for each of these genes fell within the interquartile range (IQR, Q1-Q3) indicated by the box. However, some values did fall

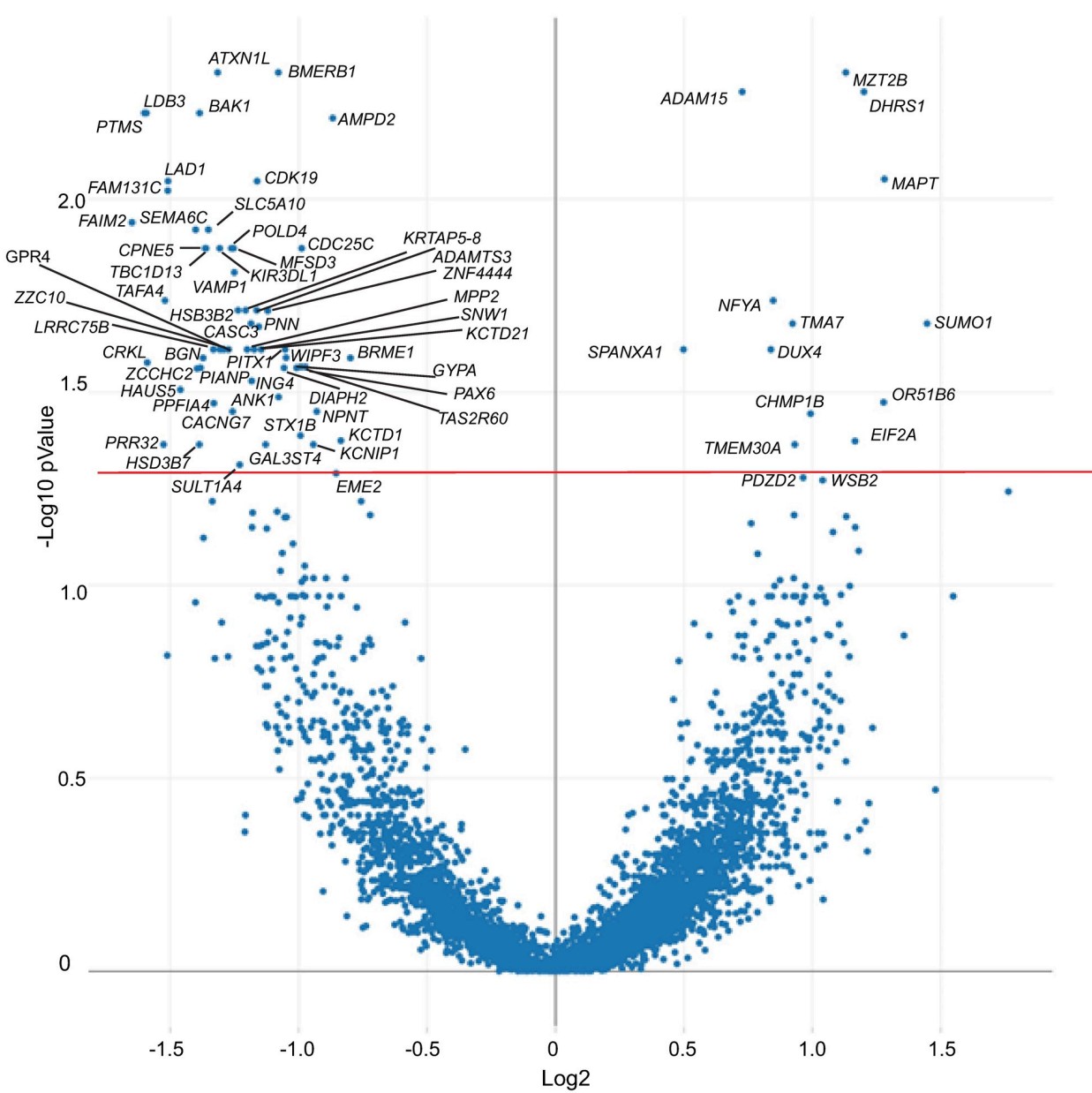

**Fig 2. Volcano plot of differentially expressed genes.** Plot shows differential expression of genes in high and low outflow areas. Each dot represents a specific gene. The solid horizontal red line shows the cutoff for genes with an adjusted p value <0.05. Only statistically significant genes are labeled.

outside the IQR. These values, which were found within the lower or higher quartile ranges, indicated that there is some variability in the expression of these genes and that their expression is not uniformly distributed across the low or high outflow regions in the TM. The whisker box-plots of *ADAM15* and *LDB3* further demonstrated the variability in expression within segmental regions of outflow. For instance, the *ADAM15* data were skewed because most of the high outflow regions had high levels of *ADAM15* expression, but three regions fell in the lower quartile range outside the IQR. The whisker boxplot for *ADAM15* in the low outflow regions showed the opposite result. Most of the low flow regions showed low levels of ADAM15 expression, but two regions had higher levels of expression outside the IQR. The whisker boxplot for

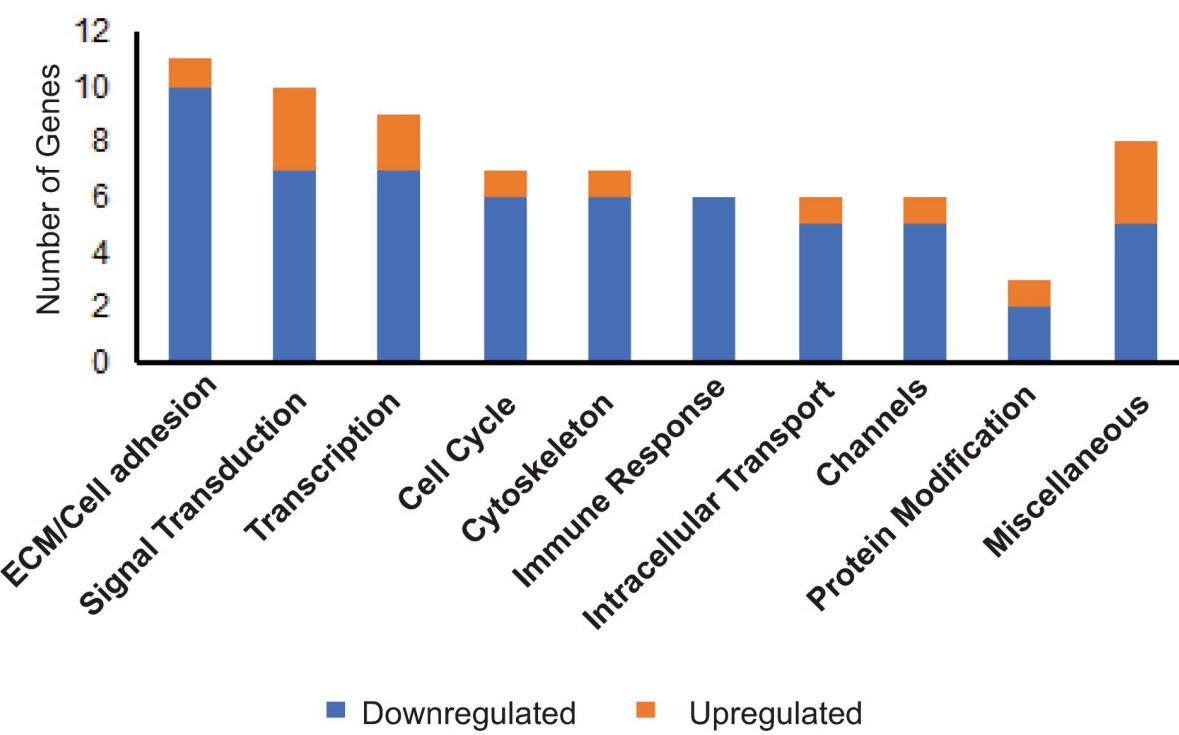

**Fig 3. Biological categories of genes exhibiting differential expression in high outflow and low outflow regions of the TM.** Biological processes were identified through gene ontology enrichment analysis using the following databases: DAVID Bioinformatics Resources, Gene Cards and UniProt.

**Table 1. ECM/cell adhesion genes differentially expressed compared to the high outflow regions.** Unless otherwise noted with a citation, the function of genes was determined using GeneCards®, the human gene database or UniProtKB/Swiss site.

| Gene | Protein Name | Function | Log2 FC | Adj p value |
|---|---|---|---|---|
| ADAM15 | ADAM metallopeptidase domain 15 | Metalloproteinase that binds β1 and β3 integrins and cleaves E-and N-cadherins and collagens | +0.7 | 0.005 |
| SPANXA1 | Sperm protein associated with the nucleus on the chromosome, family member A1 | Suppresses EMT and upregulates E-cadherin expression [38] | +0.5 | 0.02 |
| NPNT | Nephronectin | α8β1 integrin ligand | -1.0 | 0.04 |
| GYPA | Glycophorin A | Pathogen receptor, impairs cell adhesion, may interact with integrins [39, 40] | -1.0 | 0.03 |
| PNN | Pinin | Desmosome protein | -1.2 | 0.02 |
| ADAMTS3 | ADAM Metallopeptidase with Thrombospondin Type 1 Motif 3 | Disintegrin and metalloproteinase; regulates type II collagen fibrillogenesis, cleaves VEGF-C [41] | -1.2 | 0.02 |
| MPP2 | MAGUK p55 scaffold protein 2 | Organizes synapsis signaling with cell adhesion molecules | -1.2 | 0.03 |
| GPR4 | G protein-coupled receptor 4 | Increases vascular permeability, VCAM mediated cell-cell adhesion; regulates focal adhesion dynamics. Pro-inflammatory receptor. | -1.3 | 0.02 |
| PPFIA4 | PTPRF interacting protein alpha 4 (Liprin-alpha) | Phosphatase; regulates disassembly of focal adhesions | -1.3 | 0.03 |
| BGN | Biglycan | Small leucine-rich proteoglycan; crosslinks collagen fibrils and a DAMP for TLR4 | -1.4 | 0.03 |
| CRKL | CRK like proto-oncogene | Adaptor protein mediates focal adhesion formation, growth factor, integrin and cytokine signaling [42] | -1.6 | 0.03 |
| LDB3 | PDLIM6 (ZASP) | Stabilizes structural integrity of sarcomere during contraction; binds α-actinin; regulates α5β1 integrin activation [43] | -1.6 | 0.006 |

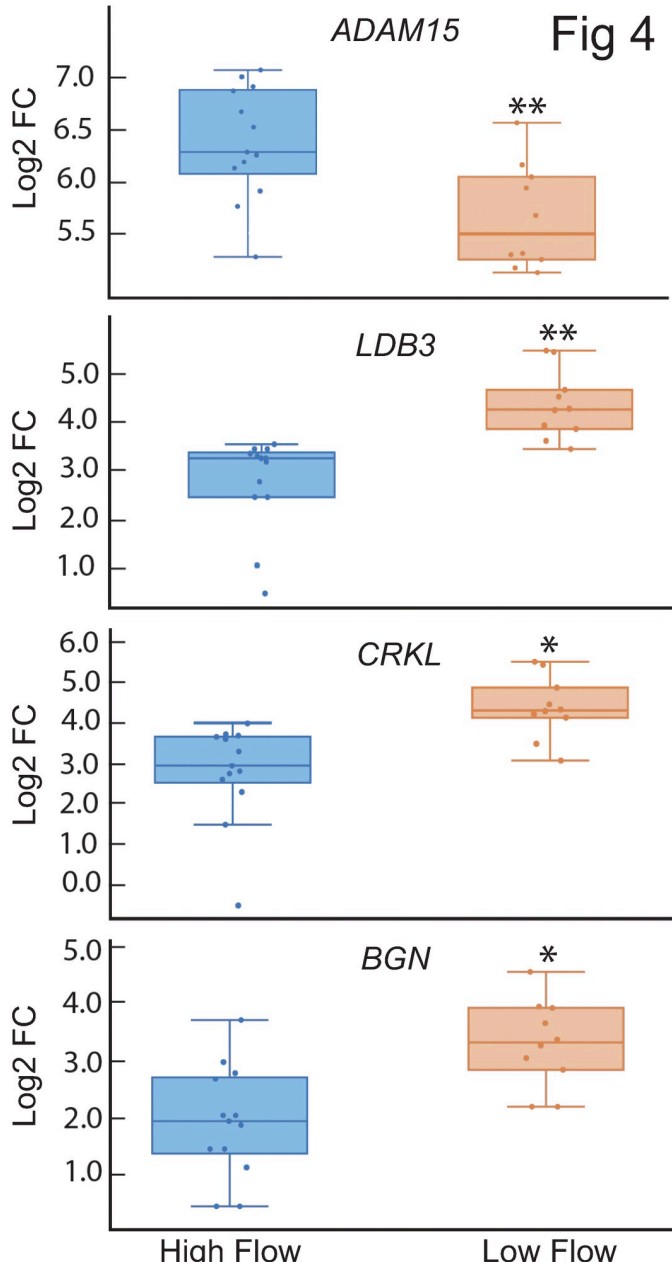

**Fig 4. Box plots showing distribution of *ADAM15*, *LDB3*, *CRKL* and *BGN* gene expression in high and low outflow regions.** Each dot represents the log2FC of the gene in one high or low outflow region. The horizontal bar represents the median of the log2FC for that gene within the interquartile range (IQR). The whiskers show the minimum and maximum log2FC change of that gene and its variability in comparison to the IQR as well as outliers. Points seen outside the box, but within the whiskers, represent data points that fall within the min (Q1-1.5* IQR) and max (Q3 + 1.5* IQR). Only data points outside the whiskers are considered outliers.

*LDB3* showed a similar distribution in low and high outflow regions. Together these data indicate the variability in gene expression within low or high outflow regions.

We did see that the data showed two outliers for *LDB3* and one outlier for *CRKL* in the high outflow segments. Further analysis showed that these outliers were from the same donor

(S1 Table). It is possible that this donor had fewer cells in those outflow regions which could account for the lower values. However, since neither *ADAM15* nor *BGN* contained outliers from these same donor tissue, this explanation is unlikely. Rather, the outliers seemed to be specific to *LDB3* and *CRKL* expression.

Immunofluorescence microscopy was then used to validate the protein expression of these genes in the TM. Out of these four proteins, only antibodies to ADAM15 were found to be available to label paraffin embedded tissues. As shown in Fig 5, this immunolabeling study showed that ADAM15 could be found in the TM. Labeling was detected in the trabecular beams, the JCT and the IW of Schlemm's Canal and appeared to be more highly expressed in these areas in high flow regions. To validate this, quantitative analysis of ADAM15 integrated density was done using Image J. This analysis showed that ADAM15 was more highly expressed in the JCT and IW of Schlemm's canal in high outflow areas compared to low out-flow areas (p<0.05). It also appeared higher in the OW of Schlemm's canal in high outflow areas, but this difference was not statistically significantly different (p>0.2).

The next category of biological processes that appeared to have genes differentially expressed was signal transduction (Fig 3). Genes found in these categories are listed in Table 2. Interestingly, half of the genes involved in signal transduction events were involved in either Wnt signaling (*KCTD1*, *CDC25C*, *BGN*, *FAIM2* and *PTMS*), TGFβ signaling (*BGN*, *PTMS*) or glucocorticoid (*CXCL1*, *PTMS*) signaling all of which are signaling pathways known to be associated with controlling outflow facility and intraocular pressure [44–48].

The third category of biological processes that had differentially expressed genes was transcription (Fig 3). As shown in Table 3, only two transcription factors were upregulated in high outflow regions. Both *NFYA* and *DUX4* showed a fold change greater than 1.7 (log2FC = 0.8), and to the best of our knowledge, neither gene has been reported to be involved in segmental outflow. All the other transcription factors were downregulated. Among the transcription factors that were differentially downregulated in high outflow by 0.4-fold was *FOSL1* (log2FC = -1.3) and *ATXN1L* (log2FC = -1.3). *FOSL1* is a subunit of the AP-1 transcription factor and has been reported to be involved in epithelial-mesenchymal transition (EMT) and Wnt signaling [56, 57], both pathways associated with IOP [48]. *SNW1*, which is involved in TGFβ [58] and Wnt signaling [59], was also detected. *PAX6* (log2FC = -1.0) was also found to be downregulated in high flow regions. Interestingly, *PAX6* has previously been associated with glaucoma and *FOSL1*, which is associated with fibrosis, were both previously identified in an earlier study exploring the effect of αvβ3 integrin on the transcriptome of TM cells [60].

Finally, another category of note included genes associated with the cytoskeleton (S2 Table). As before, the majority of genes, with the exception of one gene (*MAPT*), were downregulated in high outflow regions compared to low outflow regions. *MAPT*, which encodes for the protein Tau, showed a log2FC = 1.3 or a 2.4-fold increase in expression in high outflow regions. Genes associated with the cytoskeleton that were differentially downregulated included *LDB*3 as discussed above and *LAD1* (log2FC = -1.6). *LAD1* encodes for the protein ladinin-1 which binds to filamin-bundled actin filaments. Ladinin-1 has only recently been found and studies suggest that it may play a role in the depolymerization of F-actin filaments via its association with filamin A [66]. Its association with filamin had also been shown to affect signaling and transcriptional networks and depletion of it causes slower rates of proliferation. All other genes identified in this study and the biological processes that they are associated with are listed in S2 Table.

Interestingly, none of the genes for ECM proteins (i.e *SPARC*, *VCAN*, *THBS2*, *FN1*, *DCN)* previously associated with affecting outflow facility [20, 21, 23, 25, 67, 68] were found to be statistically different between high and low outflow regions of the TM in this study (Fig 6). Nor did we see any statistical differences in the various integrins expressed in the low outflow

# Fig 5

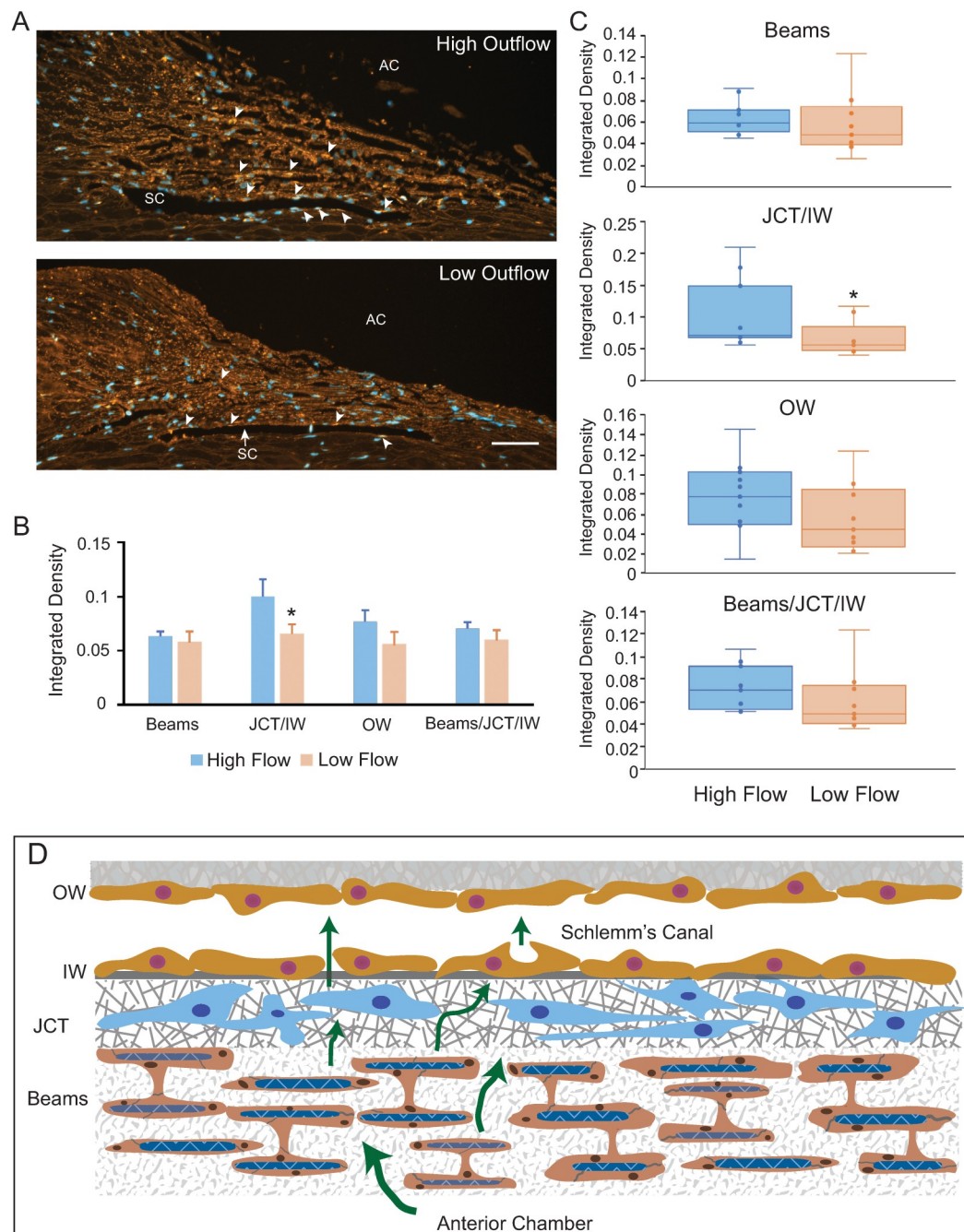

**Fig 5. Localization of ADAM15 in high and low outflow segments of TM.** (A) ADAM15 labeling (red) in high outflow regions and low outflow regions. AC, anterior chamber, SC, Schlemm's canal, Bar = 50μm. Arrowheads show ADAM15 in the beams and along Schlemm's canal. Sections were labeled with a rabbit monoclonal anti-ADAM15 antibody which was detected using Alexa 546-conjugated goat anti-rabbit IgG. Nuclei were labeled with Hoechst 33342 (blue). (B) Quantification of labeling along beams versus JCT/IW, OW and beams/JCT/IW in low and high outflow regions. * p<0.05 C) Box plots showing distribution of ADAM15 labeling in high and low outflow regions. Each dot represents the integrated density in one high or low outflow region. The horizontal bar represents the median of the integrated density. (D) Schematic of the trabecular meshwork/

Schlemm's canal outflow pathway. Aqueous humor (green arrows) flows through the trabecular beams and the JCT. It then crosses the IW of Schlemm's Canal to exit either paracellularly or transcellularly into the lumen of Schlemm's Canal and into the OW of Schlemm's Canal. The beams consist of a collagenous/elastic matrix surrounded by endothelial-like cells. The beams are connected to each other by cytoplasmic extensions between the cells surrounding the beams. IW, inner wall; OW outer wall, JCT, Juxtacanalicular region.

versus high outflow regions of the TM (Fig 7) that are thought to regulate IOP [29, 69–71]. However, boxplot analysis of the distribution of the *DCN* and *ITGA8* integrin subunits did show an enrichment of these genes in high outflow regions compared to low outflow regions. The log2FC of these genes in each outflow segment is listed in S3 Table.

**Table 2. Genes involved in signal transduction events.** Unless otherwise noted with a citation, the function of genes was determined using GeneCards®, the human gene database or UniProtKB/Swiss site.

| Gene | Protein Name | Function | Log2 FC | Adj p value |
|------|-------------|----------|---------|-------------|
| *CXCL1* | C-X-C Motif Chemokine Ligand 1 | Glucocorticoid regulation; inflammation | +1.8 | 0.05 |
| *OR51B6* | Olfactory Receptor Family 51 Subfamily B Member 6 | G-protein receptor; transduction of odorant signals | +1.2 | 0.03 |
| *PDZD2* | PDZ Domain Containing 2 | ER protein; IL-16 similarity | +1.0 | 0.05 |
| *KCTD1* | K2+ Channel Tetramerization Domain Containing 1 | Modulation of Wnt signaling; enhances ubiquitination & degradation of β-catenin; represses transcriptional activity of AP-2 | -0.8 | 0.04 |
| *CDC25C* | Cell Division Cycle 25C | Mitosis, Wnt signaling pathway [49] | -1.0 | 0.01 |
| *TAS2R60* | Taste 2 Receptor Member 60 | G-protein coupled taste receptor; mediates PLC-beta-2 activation and gating of TRPM5 | -1.0 | 0.0002 |
| *SEMA6C* | Semaphorin 6C | Involved in neuromuscular communication [50], retinal development [51]; the AKT/GSK3/β-catenin/cyclin D1 pathway [52] | -1.4 | 0.01 |
| *BGN* | Biglycan | Wnt and TGFβ signaling, collagen crosslinking | -1.4 | 0.03 |
| *TAFA4* | TAFA chemokine like family member 4 | Regulates immune or nervous cells in brain; promotes phagocytosis or ROS release in macrophages | -1.5 | 0.02 |
| *FAIM2* | Fas Apoptotic Inhibitory Molecule 2 | Regulates apoptosis and activates Wnt pathway [53] | -1.6 | 0.01 |
| *PTMS* | Parathymosin (MTI-II) | Restores BMP4 signaling [54]; glucocorticoid co-receptor [55] | -1.6 | 0.006 |

**Table 3. Genes involved in transcription.** Unless otherwise noted with a citation, the function of genes was determined using GeneCards®, the human gene database or UniProtKB/Swiss site.

| Gene | Protein Name | Function | Log2 FC | Adj p value |
|------|-------------|----------|---------|-------------|
| *NFYA* | Nuclear transcription factor Y subunit alpha | CCAAT binding transcription factor; regulates lipid metabolism and gluconeogenesis [61] | +0.8 | 0.01 |
| *DUX4* | Double homeobox 4 | Transcriptional activator of PITX1, may regulate miRNA expression | +0.8 | 0.02 |
| *PAX6* | Paired box 6 | Eye development; regulates VEGF, α4, and α5 integrin expression [62] | -1.0 | 0.03 |
| *PITX1* | Paired like homeodomain 1 | Transcription factor; regulates apoptosis-related genes (ZCCHH10 and htert) and hind limb development, [63, 64] | -1.1 | 0.02 |
| *ZNF444* | Zinc finger protein 444 | Activates transcription of a scavenger receptor gene involved in the degradation of Ac-LDL | -1.1 | 0.02 |
| *CASC3* | CASC3 exon junction complex subunit; | Component of spliceosome | -1.2 | 0.02 |
| *SNW1* | SNW domain containing 1 | Splicing and transcription, modulates TGFβ mediated transcription via Smads, coactivator that enhances glucocorticoid-mediated gene expression. | -1.2 | 0.02 |
| *ATXN1L* | Ataxin 1 like | Regulates transcription; represses notch signaling | -1.3 | 0.005 |
| *FOSL1* | FOS like 1 | AP-1 transcription factor subunit; promotes EMT and Wnt signaling [65] | -1.3 | 0.02 |

## Fig 6

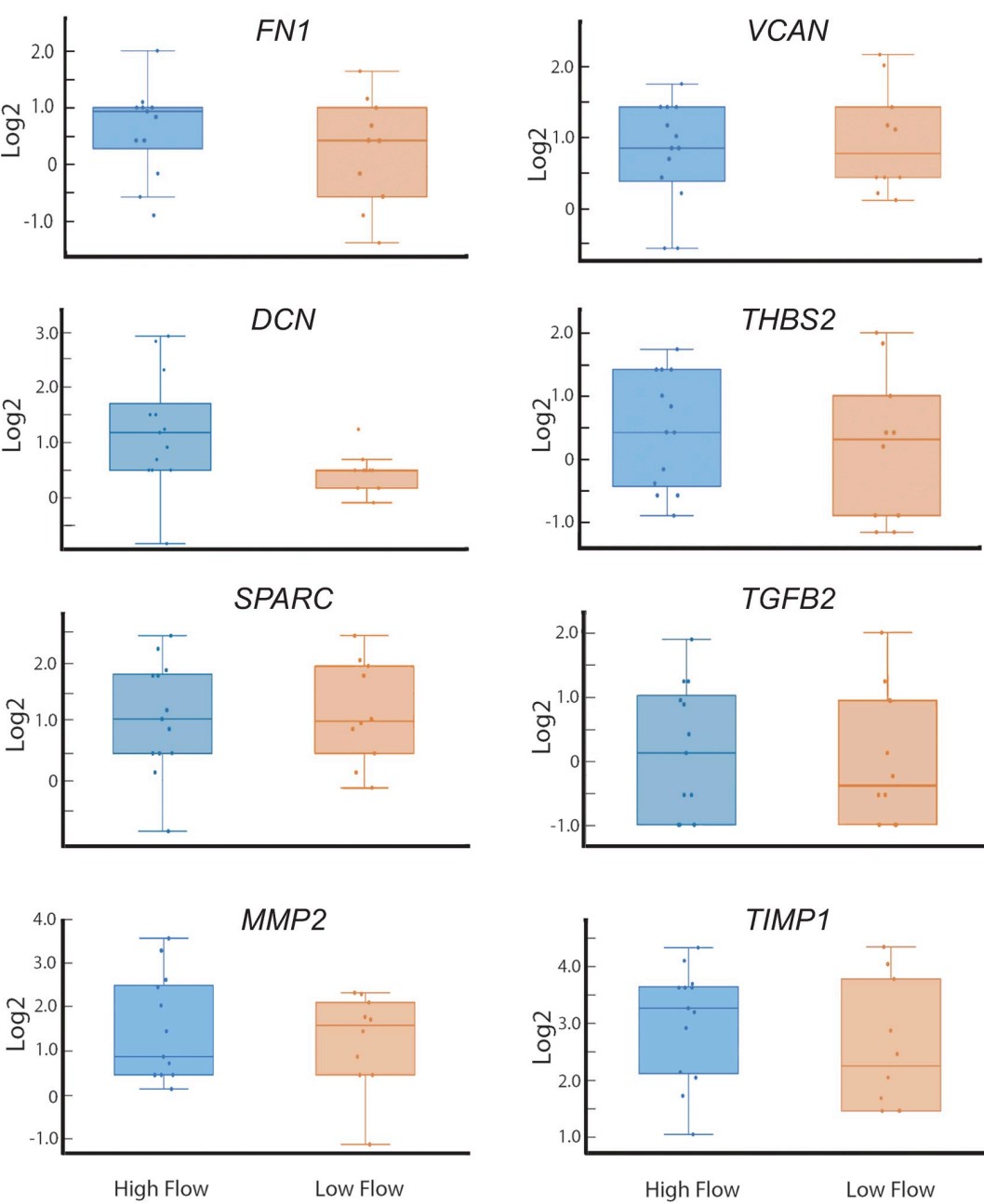

**Fig 6. Box plots showing distribution of various ECM genes and *TGF*β2 in high and low outflow regions.** Each dot represents the log2FC of the gene in one high or low outflow region. *DCN* was the only gene which showed a preferential distribution to the high outflow region of the TM. The horizontal bar represents the median of the log2FC for that gene within the IQR. The whiskers show the minimum and maximum log2FC change of that gene and its variability in comparison to the IQR as well as outliers. Points seen outside the box but within the whiskers represent data points that fall within the min (Q1- 1.5* IQR) and max (Q3 + 1.5* IQR). Only data points outside the whiskers are considered outliers. *FN1*, fibronectin; *VCAN*, versican, *DCN*, decorin; *THBS2*, thrombospondin 2; *MMP2*, matrix metalloproteinase 2; *TGFB2*, transforming growth factor 2; *TIMP1*, TIMP metallopeptidase inhibitor 1.

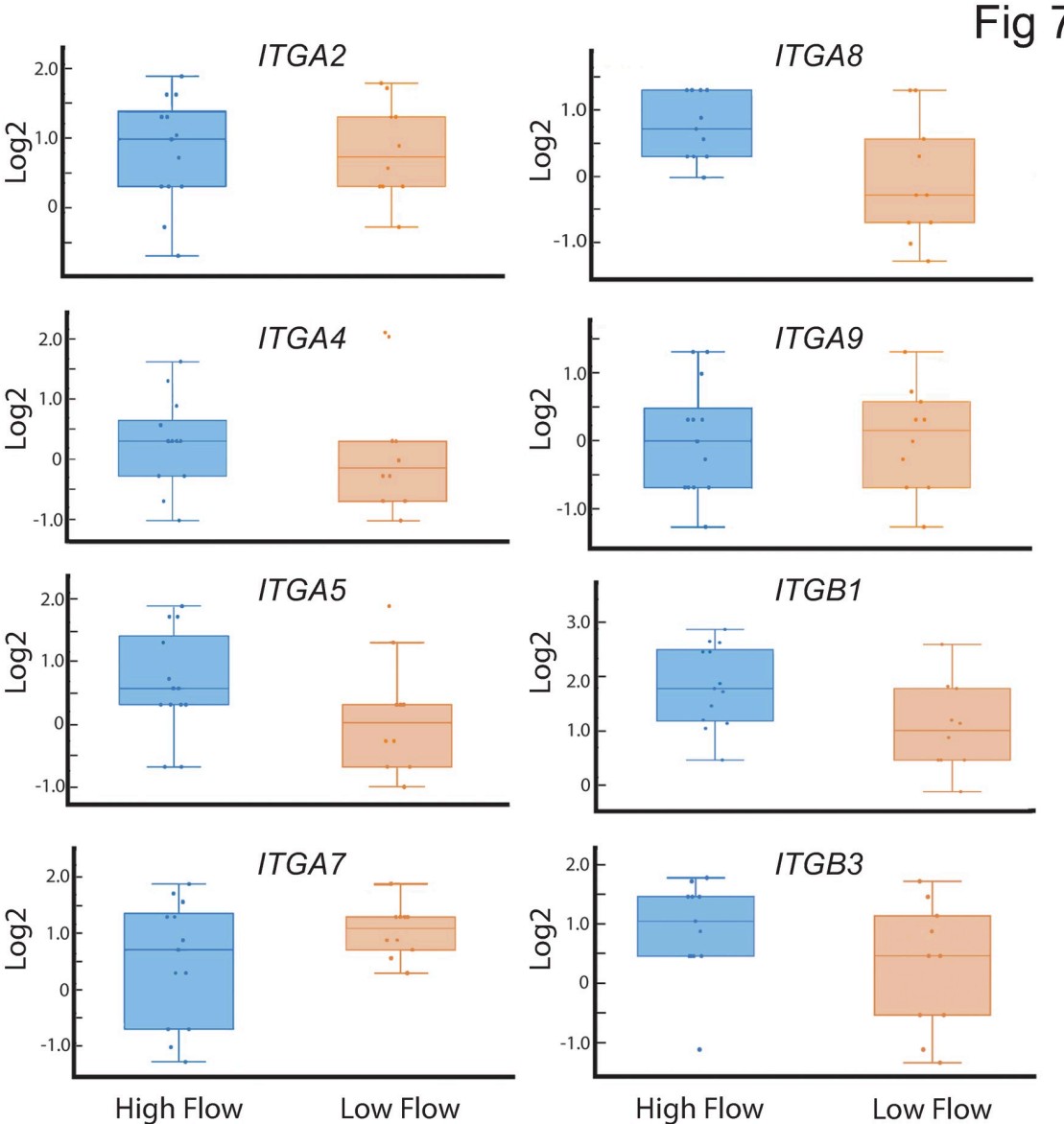

**Fig 7. Box plots showing distribution of various integrin genes in high and low outflow regions.** Each dot represents the log2FC of the gene in one high or low outflow region. Of all these integrins, only the *ITGA8* showed enrichment in the high outflow regions of the TM. The horizontal bar represents the median of the log2FC for that gene within the IQR. The whiskers show the minimum and maximum log2 FC change of that gene and its variability in comparison to the IQR as well as outliers. Points seen outside the box but within the whiskers represent data points that fall within the min (Q1-1.5* IQR) and max (Q3 + 1.5* IQR). Only data points outside the whiskers are considered outliers. *ITGA2*, α2 integrin; *ITGA4*, α4 integrin; *ITGA5*, α5 integrin; *ITGA7*, α7 integrin; *ITGA8*, α8 integrin; *ITGA9*, α9 integrin; *ITGB1*, β1 integrin; *ITGB3*, β3 integrin.

## Discussion

Using digital spatial profiling of the transcriptome of high and low outflow regions of human anterior segments, this study identified several previously unidentified genes that could be associated with segmental outflow. Many of the genes identified were not the major structural ECM proteins in the TM ECM. Rather the genes identified in this study regulated cell adhesion and/or the organization of the ECM supporting the idea that ECM organization is the critical mediator for segmental outflow [13, 31, 67].

Interestingly, the fold changes in expression of these genes were relatively small suggesting that segmental outflow is unlikely to be regulated by substantial changes in the expression of ECM genes. Previous proteomic studies [30] support this idea as the differential expression of ECM proteins in low and high outflow regions could only be detected when anterior segments were exposed to elevated pressure even though segmental outflow was detected at physiological pressure. This suggests that major changes in gene and protein expression are unlikely to account for segmental flow at physiological pressures and it is the activity or structural arrangement of ECM proteins that probably account for segmental outflow.

One of the genes identified in this study was *ADAM15*. *ADAM15* is a member of the disintegrin-metalloproteinase family and is best known for its role in cartilage homeostasis and tissue remodeling in vascular and cardiac tissues [72–74]. It has recently been suggested to orchestrate a novel pro-inflammatory mechanosensing pathway in synovial fibroblasts [75] and to play a role in regulating TGFβ signaling in cartilage [76]. Furthermore, it may modulate the activity of SLC26A2, which is a sulfate transporter responsible for maintaining adequate sulfation of proteoglycans [77]. This latter activity is intriguing since glycosaminoglycan sulfation plays an important role in the TM ECM [78–80], and it can affect the assembly of the fibronectin matrix [81], BMP-2 and TGFβ1 bioavailability and modulate signals from cell–cell and cell–ECM interactions [82]. ADAM15 also binds α9β1 and αvβ3 integrins [83, 84] and regulates αvβ3 and α7β1 integrin signaling [85, 86]. It also acts as a mechanosensory protein associated with TRPV4 Ca2+ channels [75, 87] and is upregulated by shear stress [87]. Thus, in conjugation with integrins, an ADAM15/integrin signaling complex should be able to sense elevated IOP and changes to the biomechanical environment, thereby transmitting mechanical signals intracellularly.

Together, the signaling and proteolytic activities of ADAM15, and its known association with a number of proteins/process relevant to controlling the movement of aqueous humor through the TM/SC, strongly suggest that ADAM15 can play a vital role in regulating outflow facility and the permeability of Schlemm's canal. Whether it is involved in glaucoma is still unclear. However, ADAM15 was shown to be a binding partner and target for the podosomal-adaptor protein SH3PXD2B that is mutated in the *nee* mouse model of glaucoma [88, 89]. Podosome-like structures control focal ECM turnover in TM cells [90]. Thus, any changes in ADAM15 expression could affect the podosomal degradative activity thereby affecting aqueous humor outflow through the TM [90]. Further studies examining ADAM15 levels in glaucomatous cells and tissues are clearly warranted.

Notably, previous studies have shown that another member of this family, ADAMTS4, affected outflow in human anterior segment perfusion culture [91], and perfusion of recombinant ADAMTS4 into either human or porcine anterior segments significantly increased pressure. Levels of ADAMTS4 expression were also elevated in the JCT region of the TM when anterior segments were perfused at increased pressure. Hence *ADAM15* and this family of proteins has the potential to create a microenvironment critical for TM homeostasis.

One of the striking findings of this study was the observation that none of the ECM proteins [20, 21, 23, 25, 67, 68] and integrins [29, 69–71] previously identified to affect IOP or outflow showed any significant differences in gene expression between low and high outflow regions. As shown in the boxplots for these genes and S3 Table, there was a lot of variation in the expression of genes within a high or low flow region that could not be attributed to the donor eye. For instance, in the box plot showing fibronectin expression in the high outflow regions, one particular high outflow region had the highest level of expression, while another high outflow region from the same eye had the lowest level of fibronectin expression. It should be noted, however, that this does not negate the importance of these genes in outflow facility. It simply means that differences in gene expression does not necessarily explain their

importance. Clearly, the activity of the protein [95–97] and any post-translation modifications [17, 37] as well as interactions with neighboring proteins in signaling complexes [96, 98] could be driving their role in outflow facility.

Another explanation for why a differential expression of these ECM genes was not detected could be due to the approach used to identify genes, i.e. a digital profiling technique versus isolation of RNA from dissected tissue. In addition, many of those ECM proteins were previously identified by overexpressing or knocking out a protein. In this scenario, however, the effect on segmental outflow and/or IOP is just as likely to be due to altering the 3-D structural function of the ECM as is the change in expression of one protein. Studies using artificial collagen scaffolds support this idea and show that the pore architecture of a collagen gel and glycosaminoglycan composition affects human TM cell behavior [80, 92] while the physical attributes of type I collagen fibrils in gels affects cell adhesion, integrin activation and contractility [93, 94]. This suggests that the local microarchitecture of the ECM in high and low outflow areas, and not differences in ECM protein expression, could be just as critical for regulating segmental outflow as protein levels.

Although we didn't see a differential expression of *DCN* and *ITGA8* that was statistically significant under the statistical parameters used, it is interesting that we did see an enrichment of the *DCN* gene expression in high outflow regions. Previous studies have shown that decorin expression is associated with lower IOP in a rat model of glaucoma (68) and decorin-deficient mice exhibited a higher IOP [26]. Decorin, a known antagonist of TGFβ signaling, could potentially affect outflow by attenuating the expression of TGFβ2 and target genes of the TGFβ signaling pathway [26]. The enrichment of *ITGA8* in high outflow regions is also an interesting finding. *ITGA8* is associated with vascular smooth muscle cells [95, 96] or cells that have contractile functions. This suggests that it could be involved in mediating the contractile properties of TM and/or SC cells. In vitro studies in mesenchymal cells have shown that a lack of *ITGA8* causes a loss of actin stress fibers and reorganization of the cytoskeleton [97] which could result in reduced contractility of the cell. These findings are supported by data obtained from vascular smooth muscle cells, where *ITGA8* expression promotes the contractile phenotype [98].

Interestingly, *NPNT*, which is reported to be a ligand for *ITGA8* [99] was found to be significantly decreased in high outflow regions (Table 1). This suggests that interactions between *NPNT* and *ITGA8* could be responsible for regulating outflow. For instance, changes in *NPNT* expression could induce *ITGA8* activity that may reduce outflow by controlling the contractility of TM cells.

In summary, this study shows that digital spatial profiling is a useful approach to study genes that affect outflow facility, and it has identified a number of novel, differentially expressed genes whose expression could affect segmental outflow. Many of these genes are involved in regulating cell adhesion, ECM formation and/or Wnt/TGFβ signaling suggesting that segmental outflow is likely to be regulated by the activity of the ECM and cell adhesion molecules in the TM. Clearly, additional studies examining the expression and activity of the proteins expressed by these genes are warranted.

## Supporting information

**S1 Table. Log2FC values for each segment in high and low outflow regions of TM.**
(DOCX)

**S2 Table. Genes differentially expressed in high outflow regions of TM.**
(DOCX)

**S3 Table. Log2FC values of ECM proteins and integrins for each segment in high and low outflow regions of TM.**
(DOCX)

**S1 File.**
(PDF)

## Acknowledgments

The authors thank the University of Wisconsin Translational Research Initiatives in Pathology laboratory (TRIP) for their assistance with these studies.

## Author Contributions

**Conceptualization:** Kate E. Keller, Donna M. Peters.

**Data curation:** Jennifer A. Faralli, Kate E. Keller, Donna M. Peters.

**Formal analysis:** Jennifer A. Faralli, Yong-Feng Yang, Donna M. Peters.

**Funding acquisition:** Kate E. Keller, Donna M. Peters.

**Investigation:** Jennifer A. Faralli, Mark S. Filla, Yong-Feng Yang, Ying Ying Sun.

**Methodology:** Kate E. Keller, Donna M. Peters.

**Project administration:** Donna M. Peters.

**Resources:** Kate E. Keller, Donna M. Peters.

**Supervision:** Kate E. Keller, Donna M. Peters.

**Validation:** Jennifer A. Faralli, Mark S. Filla.

**Visualization:** Donna M. Peters.

**Writing – original draft:** Donna M. Peters.

**Writing – review & editing:** Jennifer A. Faralli, Mark S. Filla, Kassidy Johns, Kate E. Keller.

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
