## [Decision Letter · Decision Letter 0]

5 Dec 2023

PONE-D-23-31232Digital spatial profiling of segmental outflow regions in trabecular meshwork reveals a role for ADAM15PLOS ONE

Dear Dr. Peters,

Thank you for submitting your manuscript to PLOS ONE. After careful consideration, we feel that it has merit but does not fully meet PLOS ONE’s publication criteria as it currently stands. Therefore, we invite you to submit a revised version of the manuscript that addresses the points raised during the review process.

 Please submit your revised manuscript by Jan 19 2024 11:59PM. If you will need more time than this to complete your revisions, please reply to this message or contact the journal office at plosone@plos.org. Please include the following items when submitting your revised manuscript:A rebuttal letter that responds to each point raised by the academic editor and reviewer(s). You should upload this letter as a separate file labeled 'Response to Reviewers'.A marked-up copy of your manuscript that highlights changes made to the original version. You should upload this as a separate file labeled 'Revised Manuscript with Track Changes'.An unmarked version of your revised paper without tracked changes. You should upload this as a separate file labeled 'Manuscript'.If applicable, we recommend that you deposit your laboratory protocols in protocols.io to enhance the reproducibility of your results. Protocols.io assigns your protocol its own identifier (DOI) so that it can be cited independently in the future. For instructions see: https://journals.plos.org/plosone/s/submission-guidelines#loc-laboratory-protocols. Additionally, PLOS ONE offers an option for publishing peer-reviewed Lab Protocol articles, which describe protocols hosted on protocols.io. Read more information on sharing protocols at https://plos.org/protocols?utm_medium=editorial-email&utm_source=authorletters&utm_campaign=protocols.

We look forward to receiving your revised manuscript.

Kind regards,

Paloma B. Liton, PhD

Academic Editor

PLOS ONE

 “This study was supported by grants EY017006, EY032905  (DMP) and EY032590, EY019643 (KK) and Core grants P30 EY01665 (UW-Dept. of Ophthalmology) and P30 EY010572 (Dept. of Ophthalmology and Casey Eye Institute).”

Reviewers' comments:

Reviewer's Responses to Questions

**Comments to the Author**

1. Is the manuscript technically sound, and do the data support the conclusions?

Reviewer #1: Partly

Reviewer #2: Yes

2. Has the statistical analysis been performed appropriately and rigorously? 

Reviewer #1: Yes

Reviewer #2: Yes

3. Have the authors made all data underlying the findings in their manuscript fully available?

Reviewer #1: Yes

Reviewer #2: Yes

4. Is the manuscript presented in an intelligible fashion and written in standard English?

Reviewer #1: Yes

Reviewer #2: Yes

5. Review Comments to the Author

Reviewer #1: Overall, Faralli et al have conducted a very interesting study further investigating ECM changes in segmental flow. We currently have minimal understanding of why outflow is segmental and how high and low flow regions change and so expanding our knowledge in this area is vital. The study shows how high/low segmental flow regions may be due to small changes in gene expression of genes that regulate ECM activity and cell adhesion.

Comments:

Line 113: the authors reference Johnson and Tschumper, 1987 for a flow rate of 6ul/min however the referenced study states that a normal human flow rate is ~2.44ul/min, and that when they perfused at 5ul/min they saw IOP at ~43 mmHg which is not consistent with the data shown by the authors (8.8mmHg and 6ul/min). The indicated pressure and flow rate give a high facility measurement (~0.7ul/min/mmHg) for perfusion studies in human donor eyes. Would this affect the segmental flow regions? Can the authors add a supplementary figure showing the perfusion trace for these studies for reference? Were there differences in the C of each donor under these conditions? It is important to see this data to consider how differences in the outflow function of each donor eye may affect the genes (low facility vs high facility eyes may have different gene expression).

Were the medium flow regions also examined as part of the study? The methods indicate they were also isolated but the data is not included in the study.

Line 218-222: the authors discuss the presence of outliers within the gene expression – were these outliers found across all 3 donors? Or were they from 1 donor? It would be helpful if each point was assigned a different shape for the source donor tissue to show biological variability.

Lines 222-225: I’m not sure what the authors mean regarding ADAM15 and LDB3 in this section – are they talking about the data shown as LDB3 does not look to be higher in the low flow compared to the other genes? Or is it simply the p-value?

Throughout the manuscript, the authors switch between LBD3 and LDB3 which is very confusing.

Figure 5: Is ADAM15 the blue/white spots? If so, what is the red? Fibronectin? The authors should clarify this.

Fig 6 and lines 289-291: Did the authors look at these genes because of previous studies showing their involvement? Decorin appears to have outliers in the high flow regions that may be contributing to the lack of significance – can the authors comment on this?

Fig 7 and associated results: what is the rationale for the authors looking at these genes?

Fig 6 and 7: the results section for these figures is very minimal, as is the discussion (except for decorin) – I’m not sure what the point of showing them is. They could maybe be moved to supplementary data.

Line 350-352: The last sentence of this paragraph does not give context as to the importance of these interactions within glaucoma/OHT.

n=3 is a pretty small sample size considering biological variables, could that account for some of the unexpected results – not seeing changes in ECM genes that have been shown to change? Were the donors all the same race?

Reviewer #2: This is an interesting study to characterize the distribution of genes and proteins in the high flow versus the low flow regions of the outflow pathway. The authors have used novel technique NanoString GeoMx platform which is an unique spatial biology platform that non-destructively profiles expression of genes and proteins on tissues. This technique has been cleverly used to gain more information into what was very little understood.

Though this study is a descriptive one more than mechanistic, the study is well rounded but minor revisions in the form of discussion can help this manuscript.

What is missing but probably out of scope for this study is comparing normal versus glaucoma samples as well as studies perfused with drugs targeting the TM can help better describe the functionality of the drugs in regulating the AH outflow by modifying the genes under study.

Interestingly the data shows that subtle changes in gene expression is happening in the low flow versus high flow. Is this a cause or an efffect of the flow changes? It is not clear from the manuscript. Moreover it might be so obvious from such studies, but a discussion can help the readers and bring in novel ideas.

Another point that will help the readers is to add in the figure a cartoon depicting - the Beams, JCT/IW, OW, and Beams/JCT/IW. This will help the trainees who are entering into the field as well as non-TM scientists to learn in a better way as to what the authors are trying to depict in terms of changes.

6. PLOS authors have the option to publish the peer review history of their article (what does this mean?). If published, this will include your full peer review and any attached files.

Reviewer #1: No

Reviewer #2: No

---

## [Author Response · Author response to Decision Letter 0]

10 Jan 2024

Reviewer #1: Overall, Faralli et al have conducted a very interesting study further investigating ECM changes in segmental flow. We currently have minimal understanding of why outflow is segmental and how high and low flow regions change and so expanding our knowledge in this area is vital. The study shows how high/low segmental flow regions may be due to small changes in gene expression of genes that regulate ECM activity and cell adhesion.

Comments:

1.Line 113: the authors reference Johnson and Tschumper, 1987 for a flow rate of 6ul/min however the referenced study states that a normal human flow rate is ~2.44ul/min, and that when they perfused at 5ul/min they saw IOP at ~43 mmHg which is not consistent with the data shown by the authors (8.8mmHg and 6ul/min). The indicated pressure and flow rate give a high facility measurement (~0.7ul/min/mmHg) for perfusion studies in human donor eyes. Would this affect the segmental flow regions? Can the authors add a supplementary figure showing the perfusion trace for these studies for reference? Were there differences in the C of each donor under these conditions? It is important to see this data to consider how differences in the outflow function of each donor eye may affect the genes (low facility vs high facility eyes may have different gene expression).

The reviewer raises an interesting point, but unfortunately, we did not monitor the pressure. We normally don’t do this for short term experiments using tracers. In these experiments, flow rates were stabilized overnight at constant pressure (8.8 mmHg) and then we only perfused the anterior segments for 1 hr with the tracer. In our experience this generates an average flow rate of 1–7 μl/min, which is similar to normal physiological rate and pressures (minus episcleral venous pressure) in vivo (Keller et al., (2008) IOVS 49: 2495–2505. We have used successfully used this constant pressure perfusion system in numerous publications exploring segmental flow and feel it eliminates issues associated with donor eyes having different pressure levels. (Vranka et al (2015) PLoS One:e0122483. doi: 10.1371/journal.pone.012248; Vranka et al., (2018) IOVS 59:246-259; Staverosky et al., (2020) EER 197:108046.doii:10.1016/jexer.2020.108046; Vranka et al., EER. (2020) 190:107888. doi: 10.1016/j.exer.2019.107888).

For this reason, we have deleted reference to the Johnson and Tschumper 1987 paper and instead cite the numerous studies mentioned using constant pressure to study segmental flow. Line 116 in the methods.

The reviewer, however, raises an excellent point about whether pressure would affect the expression of these genes and that is worth investigating in future studies using this approach since other the expression of other proteins have been found to be altered by pressure. For instance, ADAMTS4 mRNA and protein levels are increased at elevated IOP (Keller et al., (2009) IOVS 50:5769-5777), Digital spatial profiling could, therefore, be a useful new approach to study how pressure would affect gene expression under low and high outflow facility. 

2. Were the medium flow regions also examined as part of the study? The methods indicate they were also isolated but the data is not included in the study.

The kit was very expensive and limited the number of samples that we could run at one time. We decided, therefore, to only look at the high versus low flow segments, since this is where we expected to see the biggest differences. We have added text in lines 138-139 explaining this in the Methods section.

3. Line 218-222: the authors discuss the presence of outliers within the gene expression – were these outliers found across all 3 donors? Or were they from 1 donor? It would be helpful if each point was assigned a different shape for the source donor tissue to show biological variability. 

The two outliers seen in the LDB3 boxplot and the one outlier seen in the CRKL boxplot are the values that fell outside the min (Q1-1.5*inter Quartile Range) and max (Q3 + 1.5* Inter Quartile Range). Although all these 3 data points did come from the same donor eye, it is impossible to know why these high outflow segments from this donor had these outliers and whether these outliers are legitimate observations. It is possible that this donor had fewer cells in those outflow segments which could account for the lower values. However, ADAM15 was highly expressed in these segments (~ log2=7) and showed the same values seen in other donor eyes, so that explanation seems unlikely. Rather these values appear to reflect variations seen throughout the various segmental outflow regions within a donor eye. 

Although we agree that assigning a different shape for the data based on the different donors would be helpful, the Nanostring software which generated these figures would not allow us to make such a change. Nor could we find a program that would graph the data that way. Instead, we have added a Supplemental Table (S1) which assigns the data obtained for LDB3, ADAM15, CRKL, and BGN to individual a particular donor eye. 

We have added text addressing this question to the results section (lines 221-240) of the manuscript. 

4. Lines 222-225: I’m not sure what the authors mean regarding ADAM15 and LDB3 in this section – are they talking about the data shown as LDB3 does not look to be higher in the low flow compared to the other genes? Or is it simply the p-value? 

We apologize for the confusion. We are referring to the variations of LDB3 or ADAM15 expression within the high or low outflow regions. We are not comparing between the high and low outflow regions. We wanted to point out that in the LDB3 and ADAM15 boxplots, the data are not symmetrical (i.e bell-shaped distribution) which we feel further emphasizes the variations seen throughout the different outflow segmental regions. 

We have tried to clarify this point within the same text addressing query #3 (lines 221-240). 

4. Throughout the manuscript, the authors switch between LBD3 and LDB3 which is very confusing. 

We apologize for the confusion. This was obviously a typo which we have corrected throughout the manuscript. 

5.Figure 5: Is ADAM15 the blue/white spots? If so, what is the red? Fibronectin? The authors should clarify this. 

We are sorry for the confusion. The blue spots are the nuclei labeled with Hoechst 33342. The red is labeling is ADAM15 labeling. Sections were then labeled with rabbit monoclonal anti-ADAM15 and labeled with a secondary Alexa 546-conjugated goat anti-rabbit IgG as we described in the methods. We have added this text (lines 263-265) to the figure legend to make it clear. 

6. Fig 6 and lines 289-291: Did the authors look at these genes because of previous studies showing their involvement? 

Yes, we thought it was important to show this negative data as well as the variation in gene expression that you would see even within the same eye. This is something rarely discussed in the field. For instance, in the box plot showing fibronectin expression in the high flow region, one high flow region had the highest level of expression and another high flow region from the same eye had the lowest level of fibronectin expression. This study is the first to statistically document this observation and it is for this reason that we thought it was important to show the data and inform the field of this issue.

We have added text about this in the results (lines 320-323) and discussion sections (lines 396-405).

7. Decorin appears to have outliers in the high flow regions that may be contributing to the lack of significance – can the authors comment on this?

With all due respect, the reviewer is incorrect. Decorin did not have any outliers in the high flow regions. The box in the plot represents data found within the 25th (Q1) and 75th (Q3) percentile. Those points seen outside the box represent data points that fell within the maximum for the Q3 interquartile + 1.5 IOR and are not considered outliers. We have added additional text (lines 329-331) in the figure legend to clarify. 

The boxplot for the low flow region, however, does show that decorin has one outlier in the low flow region that fell outside the min (Q1-1.5* Inter Quartile Range) and max (Q3 + 1.5* Inter Quartile Range). Table S3 shows the donor in which the outlier was observed. As we discussed above we do not know why these outliers exist. Clearly, additional studies are warranted to understand why outliers appear. 

7. Fig 7 and associated results: what is the rationale for the authors looking at these genes?

We thought it was important to show this negative data as we know many investigators in the field are looking at integrins because they play such a significant role in many signaling pathways. As with the ECM genes we see variation in integrin gene expression that you would see even within the same eye. It is for this reason that we thought it was important to show the data and inform the field of this issue.

8. Fig 6 and 7: the results section for these figures is very minimal, as is the discussion (except for decorin) – I’m not sure what the point of showing them is. They could maybe be moved to supplementary data.

We didn’t put it in the supplement because people don’t always look at the supplement and we knew there would be questions about what happened to these genes. We felt putting it into the body of the manuscript would increase the transparency of the data and as we mentioned above, would further highlight the variation that we saw even within the same eye. We have added more text to the results (line 320) and discussion sections (lines 394-405) about this.

9. Line 350-352: The last sentence of this paragraph does not give context as to the importance of these interactions within glaucoma/OHT.

Thank you for the comment. We have edited the text in this paragraph and added the following paragraph to give more relevance to glaucoma/OHT. The paragraph says: “Together, the signaling and proteolytic activities of ADAM15, and its known association with a number of proteins/processes relevant to controlling the movement of aqueous humor through the TM/SC, strongly suggest that ADAM15 can play a vital role in outflow facility and the permeability of Schlemm’s canal. Whether it is involved in glaucoma is still unclear. However, ADAM15 was shown to be a binding partner and target for the podosomal-adaptor protein SH3PXD2B mutated in the nee mouse model of glaucoma (Mao et al., Mamm Genome 2009 20:462-475; Mao et al., 2011. IOVS 52:2679-2688). Podosome-like structures control focal ECM turnover in TM cells. Thus, any changes in ADAM15 expression could affect the podosomal degradative activity thereby affecting aqueous humor outflow through the TM (Aga et al., 2008 IOVS 49:5353-5365). Further studies examining ADAM15 levels in glaucomatous cells and tissues are clearly warranted.” Please refer to lines 375-384 for this text.

10. n=3 is a pretty small sample size considering biological variables, could that account for some of the unexpected results – not seeing changes in ECM genes that have been shown to change? Were the donors all the same race?

Given the availability of human cadaver eyes and the cost of the assay, we limited the study to n=3. Obviously, it would have been preferable to increase the number of ‘n’s, but increasing biological replicates does not always produce cleaner data, especially when dealing with human eyes with differing genetic backgrounds and medical histories. Future studies using this approach with mouse models of ocular hypertension may prove to be more informative. Nevertheless, we have seen statistically significant differences in human eyes illustrating that this new technique can be applied to study segmental gene expression and changes in gene expression associated with glaucoma. 

One of the donors was Caucasian, but there is no race noted for the other 2 donors on the datasheet we received from the eyebank. We added this information to the methods (lines 99-100). 

Reviewer #2: This is an interesting study to characterize the distribution of genes and proteins in the high flow versus the low flow regions of the outflow pathway. The authors have used novel technique NanoString GeoMx platform which is a unique spatial biology platform that non-destructively profiles expression of genes and proteins on tissues. This technique has been cleverly used to gain more information into what was very little understood.

Though this study is a descriptive one more than mechanistic, the study is well rounded but minor revisions in the form of discussion can help this manuscript.

1. What is missing but probably out of scope for this study is comparing normal versus glaucoma samples as well as studies perfused with drugs targeting the TM can help better describe the functionality of the drugs in regulating the AH outflow by modifying the genes under study.

We agree with the reviewer that this would be an interesting study. However, for this paper we decided to focus on understanding the genes that may be enriched in high outflow versus low outflow regions. It is our feeling that having a better understanding of what could be influencing normal outflow can be helpful for understanding the changes found in glaucomatous patients. As such we feel that this study makes a valuable contribution to the field as it shows not only the feasibility of using this approach, but it demonstrates the variability that we are seeing at the gene level as we now discuss in the paper. 

2. Interestingly the data shows that subtle changes in gene expression is happening in the low flow versus high flow. Is this a cause or an effect of the flow changes? It is not clear from the manuscript. Moreover it might be so obvious from such studies, but a discussion can help the readers and bring in novel ideas. 

Based on this data, it is hard to say whether these changes are the cause or an effect of flow change. Although this is a very interesting question, we feel that to speculate on it would be beyond the focus of this manuscript. 

3. Another point that will help the readers is to add in the figure a cartoon depicting - the Beams, JCT/IW, OW, and Beams/JCT/IW. This will help the trainees who are entering into the field as well as non-TM scientists to learn in a better way as to what the authors are trying to depict in terms of changes.

As suggested by the reviewer, we have added a schematic diagram depicting the TM Beams, JCT/IW, OW, and Beams/JCT/IW into Figure 5.

---

## [Decision Letter · Decision Letter 1]

31 Jan 2024

Digital spatial profiling of segmental outflow regions in trabecular meshwork reveals a role for ADAM15

PONE-D-23-31232R1

Dear Dr. Peters,

We’re pleased to inform you that your manuscript has been judged scientifically suitable for publication and will be formally accepted for publication once it meets all outstanding technical requirements.

Kind regards,

Paloma B. Liton, PhD

Academic Editor

PLOS ONE

Additional Editor Comments (optional):

Reviewers' comments:

Reviewer's Responses to Questions

**Comments to the Author**

1. If the authors have adequately addressed your comments raised in a previous round of review and you feel that this manuscript is now acceptable for publication, you may indicate that here to bypass the “Comments to the Author” section, enter your conflict of interest statement in the “Confidential to Editor” section, and submit your "Accept" recommendation.

Reviewer #1: All comments have been addressed

Reviewer #2: All comments have been addressed

2. Is the manuscript technically sound, and do the data support the conclusions?

Reviewer #1: Yes

Reviewer #2: Yes

3. Has the statistical analysis been performed appropriately and rigorously? 

Reviewer #1: Yes

Reviewer #2: Yes

4. Have the authors made all data underlying the findings in their manuscript fully available?

Reviewer #1: Yes

Reviewer #2: Yes

5. Is the manuscript presented in an intelligible fashion and written in standard English?

Reviewer #1: Yes

Reviewer #2: Yes

6. Review Comments to the Author

Reviewer #1: (No Response)

Reviewer #2: The manuscript looks well rounded now. The authors have addressed all the comments from the reviewers.

7. PLOS authors have the option to publish the peer review history of their article (what does this mean?). If published, this will include your full peer review and any attached files.

Reviewer #1: No

Reviewer #2: No

---

## [Editor Report · Acceptance letter]

12 Feb 2024

PONE-D-23-31232R1 

PLOS ONE

Dear Dr. Peters, 

I'm pleased to inform you that your manuscript has been deemed suitable for publication in PLOS ONE. Congratulations! Your manuscript is now being handed over to our production team.

Kind regards, 

on behalf of

Dr. Paloma B. Liton 

Academic Editor

PLOS ONE